# Sepsis—A Retrospective Cohort Study of Bloodstream Infections

**DOI:** 10.3390/antibiotics9120851

**Published:** 2020-11-28

**Authors:** Biagio Santella, Veronica Folliero, Gerarda Maria Pirofalo, Enrica Serretiello, Carla Zannella, Giuseppina Moccia, Emanuela Santoro, Giuseppina Sanna, Oriana Motta, Francesco De Caro, Pasquale Pagliano, Mario Capunzo, Massimiliano Galdiero, Giovanni Boccia, Gianluigi Franci

**Affiliations:** 1Section of Microbiology and Virology, University Hospital “Luigi Vanvitelli”, 80138 Naples, Italy; bi.santella@gmail.com (B.S.); enrica.serretiello@unicampania.it (E.S.); massimiliano.galdiero@unicampania.it (M.G.); 2Department of Experimental Medicine, University of Campania “Luigi Vanvitelli”, 80138 Naples, Italy; veronica.folliero@unicampania.it (V.F.); carla.zannella@unicampania.it (C.Z.); 3Dai Dipartimento Di Igiene Sanitaria e Medicina Valutativa U.O.C. Patologia Clinica E Microbiologica, Azienda Ospedaliero-Universitaria S. Giovanni di Dio e Ruggi D’Aragona Scuola Medica Salernitana, Largo Città di Ippocrate, 84131 Salerno, Italy; gerard732000@yahoo.it (G.M.P.); fdecaro@unisa.it (F.D.C.); mcapunzo@unisa.it (M.C.); 4Department of Medicine, Surgery and Dentistry “Scuola Medica Salernitana”, University of Salerno, 84081 Baronissi, Italy; gmoccia@unisa.it (G.M.); esantoro@unisa.it (E.S.); omotta@unisa.it (O.M.); ppagliano@unisa.it (P.P.); 5Department of Biomedical Sciences, University of Cagliari, Cittadella Universitaria, Monserrato, 09042 Cagliari, Italy; g.sanna@unica.it

**Keywords:** antimicrobial sensitivity, blood culture, bloodstream infections, empiric therapy

## Abstract

Bloodstream infections (BSIs) are among the leading causes of morbidity and mortality worldwide, among infectious diseases. Local knowledge of the main bacteria involved in BSIs and their associated antibiotic susceptibility patterns is essential to rationalize the empiric antimicrobial therapy. The aim of this study was to define the incidence of infection and evaluate the antimicrobial resistance profile of the main pathogens involved in BSIs. This study enrolled patients of all ages and both sexes admitted to the University Hospital “San Giovanni di Dio e Ruggi d’Aragona”, Salerno, Italy between January 2015 to December 2019. Bacterial identification and antibiotic susceptibility testing were performed with Vitek 2. A number of 3.949 positive blood cultures were included out of 24,694 total blood cultures from 2015 to 2019. Coagulase-negative staphylococci (CoNS) were identified as the main bacteria that caused BSI (17.4%), followed by *Staphylococcus aureus* (12.3%), *Escherichia coli* (10.9%), and *Klebsiella pneumoniae* (9.4%). Gram-positive bacteria were highly resistant to Penicillin G and Oxacillin, while Gram-negative strains to Ciprofloxacin, Cefotaxime, Ceftazidime, and Amoxicillin-clavulanate. High susceptibility to Vancomycin, Linezolid, and Daptomycin was observed among Gram-positive strains. Fosfomycin showed the best performance to treatment Gram-negative BSIs. Our study found an increase in resistance to the latest generation of antibiotics over the years. This suggests an urgent need to improve antimicrobial management programs to optimize empirical therapy in BSI.

## 1. Introduction

Bloodstream infections (BSIs) represent a major cause of mortality and morbidity worldwide [1]. BSIs are widely spread all over the world with direct and indirect social and economic impacts. It is estimated that BSIs affects approximately 30 million people, causing 6 million deaths each year in the world [2,3]. European reports revealed that BSI cases are more than 1.2 million each year, with a number of deaths around 157,000 patients [4]. Based on the age group, previous morbidity, and other risk factors, the mortality rate of BSIs ranges between 4.0 and 41.5% [5,6,7,8,9,10]. Healthcare costs range between $10,000 and $20,000 per hospitalized patient [11]. BSIs are caused by the presence of live bacterial and/or fungal microorganisms in the bloodstream. These events can favor a strong inflammatory response, with alteration of some clinical and hemodynamics parameters [12]. The BSIs can be divided into primary or secondary infections [13]. In primary BSIs, microorganisms are introduced directly into the bloodstream, for example, through the use of contaminated medical devices. Secondary BSI is a bloodstream infection driven by the same organism causing infection in another host tissue [14]. Moreover, BSIs are further classified in community or hospital-acquired infections. Differences among them are due to the place and duration of infection. In particular, the first one manifests in a community or within the first 48h of admission in the hospital. In the second group, BSI occurs 48 h after admission or 3 days after hospital discharge [15]. Bacteria are the leading cause of BSIs, although also the fungi may be implicated in the emergence of this infection [16]. Previous studies detected *Escherichia coli*, *Klebsiella pneumoniae*, *Staphylococcus aureus*, and CoNS (Coagulase-negative staphylococci) strains as the most common cause of BSIs [3]. The diagnosis of BSIs is carried out through the analysis of clinical symptoms of the patient and laboratory tests [17]. The most common clinical symptoms of BSIs include (i) fever (>38 °C), (ii) chills, (iii) hypotension, and (iv) increase in white blood cell count and inflammation markers concentrations [13]. Nevertheless, the reported signs are not always present: for instance, elderly patients may develop a subdued fever or hypothermia in severe outcomes. Likewise, tachycardia and tachypnea represent the clinical signs most commonly present in critical patients. Moreover, leukopenia can be seen in severe cases, when the white blood cell is activated massively and entrapped in peripheral sites [18]. Given the complexities of clinical signs, only microbiological analysis of blood samples confirms the clinical diagnosis of sepsis [19]. Actually, blood cultures (BCs) represent the gold standard method for the diagnosis of BSIs [20]. BCs provide the identity of the pathogen and the relative pattern of antibiotic susceptibility with high sensitivity [21]. The epidemiology of BSIs differs between several countries [3,22,23,24]. These significant differences between healthcare communities require constant monitoring of local trends. The advent of antimicrobial resistance (AMR) among most bacterial pathogens causes a serious health crisis with many economic and social implications around the world [25]. AMR threatens the efficacy of antibiotics frequently used to prevent and treat BSIs. Furthermore, the lack of novel antibiotics highlights the limitations of the situation and underlines the needs of programs and actions in order to face the problem [26]. The starting point is represented by studies on bacterial etiology and antibiotic resistance profile of bacterial BSIs to improve the empirical treatment and the administration of the correct antibiotic therapy [5]. In this scenario, the current study was carried out to evaluate the bacterial pathogens involved in BSIs and their antimicrobial susceptibility pattern in patients admitted to the San Giovanni di Dio e Ruggi d’Aragona Hospital (Salerno, Italy). Knowledge about the main bacterial BSIs and related antibiotics susceptibility profile is crucial to permit the appropriate choice of antibiotic treatment, leading to a reduction in hospital stay, the cost of therapy, and mortality.

## 2. Results

### 2.1. Incidence of BSIs in Studied Patients

In the present study, 24,694 blood samples were examined. BSIs were diagnosed based on the patient’s clinical signs, biochemical parameters, and the presence of microorganisms in the blood. From 2015 to 2019, 3949 cases of BSIs were recorded (Table 1). The mean of patients with BSIs was 16.4%. Of these, 2841 (71.9%) and 1108 (28.1%) were diagnosed as primary and secondary bacteremia, respectively. Primary and secondary BSIs were included in our analysis.

Our findings showed a rather linear trend over the years, except for the year 2018, where the number of positives exceeded 22% (Figure 1).

The gender and age distributions of the patients with BSIs are reported in Figure 2 and Figure 3. Regarding gender, the BSIs rate was higher in males than in females (Figure 2). Concerning age distribution, most of the positive patients were placed in the 61–80 age group (Figure 3).

### 2.2. Isolated Bacteria

All pathogens identified over the 5 years of study, with respective incidence rates, were provided as additional data (Appendix A). Our data showed that CoNS strains were the main isolated strains in the period from 2015 to 2017 and 2019 (>15.32%). Only in 2018, *Staphylococcus aureus* was the most identified strain (18.20%). *Escherichia coli* was ranked second in 2015 and 2017, while *Staphylococcus aureus* in 2016 and 2019. *Klebsiella pneumoniae* was the least represented strain. In 2018–2019, this strain was the third most isolated strain (Figure 4).

### 2.3. Prevalence of Antimicrobial Resistance among BSI Bacteria

In the present study, the antimicrobial resistance profile of *Staphylococcus aureus*, CoNS strains, *Escherichia coli*, and *Klebsiella pneumoniae* had been analyzed. The antimicrobial resistance patterns are shown in Table 2, Table 3, Table 4 and Table 5. All isolated strains showed a high rate of resistance to the tested antibiotics. Among the Gram-positive bacteria analyzed, the most resistant species is represented by *Staphylococcus aureus*. The resistance rates for *Staphylococcus aureus* to Penicillin were higher than 84.6% in 2015–2018. It was relevant reduced in 2019 (68.9%). Resistance to Gentamicin in *Staphylococcus aureus* exhibited a relevant downward trend, ranging from 13.3 to 7.8%. The resistance to Oxacillin was detected in 229 of 515 total isolated strains of *S. aureus* (44%). Clindamycin, Erythromycin, Levofloxacin, Rifampicin, and Tetracycline fluctuated lightly, ranging from 36.7 to 35.9%, 43.3 to 42.7%, 33.3 to 27.8%, 7 to 7.8%, and 9.6 to 9.8%, respectively. The resistance rates for *S. aureus* to Vancomycin, Teicoplanin, Daptomycin, and Linezolid were lower than 7.8% but in an alarming increase (Table 2).

CoNS strains represent the most frequent Gram-positive bacteria, involved in BSIs. In this study, they include the following species; *Staphylococcus epidermidis*, *Staphylococcus haemolyticus*, and *Staphylococcus hominis*. CoNS represent the most common blood culture contaminant. In order to differentiate contamination from real bacteremia, more than two culture series were performed per patient at the same and different time by separate venipuncture. The presence of the same isolate on multiple culture blood sets allowed us to differentiate false positives from true positives. As contamination can arise during the collection phase, having reliable factors is essential for patient management and patient surveillance [27]. The resistance rates for CoNS strains to Oxacillin were higher than 77.5%. It fluctuated slightly, ranging from 81.1 to 77.8%. Furthermore, the data analysis indicated a light variation of resistance percentage of Fusidic acid, Clindamycin, and Rifampicin, passing from 28.3 to 33.1%, 62.5 to 51.4%, and 28 to 34.3%, respectively. The resistance frequency to Daptomycin, Linezolid, and Vancomycin was less than 6.9%. In this case, the increase in resistance is worrying (Table 3).

Our findings showed that among Gram-negative bacteria, *Klebsiella pneumoniae* was the most resistant strain. The resistance rates for *K. pneumoniae* to Ertapenem, Meropenem, Piperacillin/Tazobactam, and Ciprofloxacin were above 54.9% and varies slightly over the studied period. A significant increase was observed for the resistance rates to Amoxicillin/Clavulanic acid, Cefotaxime, Ceftazidime, and Fosfomycin which passed from 77.6 to 94.8%, 85.1 to 92.4%, 79.1 to 92.5%, and 22.4 to 38.2%, respectively. A slight decrease was showed in Trimethoprim/Sulfamethoxazole resistance rate, from 77.6 to 58.1% (Table 4).

The antimicrobial susceptibility data exhibited that the resistance rate for *Escherichia coli* to Amoxicillin/Clavulanic acid, Ciprofloxacin, and Trimethoprim/Sulfamethoxazole was over 50%. Resistance to third-generation cephalosporins (Cefotaxime) was found in 54.1% of the *E. coli* strain. In contrast, Ertapenem, Fosfomycin, Imipenem, Meropenem, and Tigecycline resistance were below 5%. The resistance rates for all reported antibiotics fluctuated slightly in the studied period (Table 5).

## 3. Discussion

BSIs represent a global problem that needs prompt action. Timely detection, identification, and antimicrobial susceptibility testing of causative pathogens and hospital surveillance is needed to improve BSI management. This study shows the prevalence of BSI, the incidence of pathogens causing the infection, and evaluates the sensitivity profile to the main antibiotics used in the treatment, in the period between January 2015 and December 2019. In this study period, 24,694 total blood cultures were included, of which 3949 were positive (16%). For the years examined, a rather linear incidence trend (15–16%) was calculated, with the exception of 2018, where an incidence exceeds 22%. This higher value may be justified by a lower total number of blood cultures received and by a high positivity, probably caused by the *S. aureus* strains, more frequent in the year under review. Secondary BSIs was diagnosed in 1108 patients (28%). Our data were similar to those reported by the European Center for Disease Prevention and Control, which recorded 29% of secondary bacteremia [28]. Lower frequency of secondary BSIs was observed at University Hospital of the Canary Islands in Spain (22%) [29]. In the current study, the secondary bloodstream infections were not classified according to the district of the associated infectious process. In addition, all cases of bacteremia were included in our analysis, with the aim of evaluating the incidence and trends of antibiotic resistance of the main pathogenic bacteria that cause BSI. The average incidence of BSI cases was assessed, with a response of 16.4% per year. This isolation rate is consistent with many studies in Europe and abroad [3,30]. The incidence of bacteremia, in this study, increases rapidly with increasing age and is higher in males, in accordance with other studies [31,32]. CoNS (Coagulase-negative staphylococci) (17.4%), *Staphylococcus aureus* (12.8%), *Escherichia coli* (10.9%), and *Klebsiella pneumoniae* (9.4%) were the most frequently isolated species. As in other studies, the predominance of Gram-positive pathogens was documented and the bacterial distribution patterns were consistent with those reported [30,33,34,35]. Among the Gram-positive bacteria analyzed, the most resistant species is represented by *Staphylococcus aureus*. This species showed high resistance to Oxacillin (43.7%), Erythromycin (47.8%), and Levofloxacin (38.2%). The spread of methicillin-resistant *S. aureus* (MRSA) is a major public health problem [36]. To date, the most suitable treatment for resolving infections caused by MRSA is represented by to use of Glycopeptides, in particular, Vancomycin. Overuse of this antibiotic led to the emergence of resistant strains, present in this study, with an average percentage of 3.6% in the last four years, compared to 2015, where resistant strains were absent. In recent years, new antibiotics such as Linezolid and Daptomycin have been introduced into clinical practice for the treatment of MRSA infections [37]. Furthermore, for these antibiotics, as for vancomycin, there is a resistance rate of 3.9% in the last years of the study, compared to 2015 (0%). However, in this study, CoNS were the most frequent strains involved in BSI among Gram-positive bacteria. Although CoNS may act as contaminants in some cases [38], they were carefully evaluated before being included in this study, as reported in the results section. These species showed high resistance to Clindamycin (56.1%) and Oxacillin (72.7%), while low resistance was shown for Daptomycin, Linezolid and Vancomycin. CoNS are known to be the reservoir of resistance genes; therefore, the resistances shown in this study could spread among pathogenic staphylococci such as MRSA, and increase the difficulties in treating pathogen-promoted MDR infections [39]. The main Gram-negative species causing BSI, in this study, were *Escherichia coli* and *Klebsiella pneumoniae*, with a mean incidence of 10.9% and 9.5%, respectively. Isolates of *K. pneumoniae* showed high resistance to third-generation cephalosporins (Ceftazidime and Cefotaxime), fluoroquinolones (Ciprofloxacin), and carbapenems (Meropenem and Ertapenem), with an overall resistance range between 65% and 84%. However, they showed low resistance to Colistin and Fosfomycin, with a value of around 25%, for both. In contrast, *E. coli* isolates showed much lower resistance values towards carbapenems (Ertapenem, Meropenem, and Imipenem) around 1%, although they share high resistance values to third-generation cephalosporins and fluoroquinolones. These values were higher than the European average and the national average [40]. By using broad-spectrum antibiotics, such as cephalosporin and fluoroquinolones, they have favored the colonization and spread of resistant Enterobacteriaceae, including *E. coli* [41]. In 2019 there was a relevant increase in the percentage of *K. pneumoniae* isolates to penicillin-resistant (Amoxicillin/Clavulanic Acid) from 77.6% in 2015 to 94.8% in 2019. Furthermore, the study showed an increase in the percentage of resistance to third-generation both around 92% in 2019. While resistance to carbapenems from 64.2% in 2015 to 65.6% in 2019, and to fluoroquinolones, from 86. 6% in 2015 to 89.2% in 2019, has remained fairly stable, even if very high values are recorded, compared to the European average (among 25–50%) [42,43].

## 4. Materials and Methods

### 4.1. Samples Collection

A total of 24,694 blood samples were collected from patients admitted to the University Hospital “San Giovanni di Dio e Ruggi d’Aragona” in the period between January 2015 and December 2019. Blood draw was performed in accordance with the hospital’s aseptic guidelines. The protocol required the disinfection at the collection site with 2% chlorhexidine.

### 4.2. Isolation, Identification and Antimicrobial Susceptibility Test for BSI Pathogens

A volume of 5–10 mL and 2–3 mL were inoculated in blood culture bottles for adult and pediatric patients, respectively. For pediatric patients, the survey was performed only on the aerobic bottle, while an aerobic bottle and an anaerobic bottle were used for adult patients. Blood culture samples were delivered to the Microbiology Laboratory for testing. Blood culture bottles were incubated in the automated blood culture monitoring BACTEC 9240 blood culture system (Becton Dickinson Diagnostic Instrument Systems) system. The most common incubation time for bacteria was 5 days, it was increased for slow-growing organisms. When a positive alarm occurred in the blood culture instrument, 1 drop from each bottle was plated on standard bacteriology media: Chocolate agar, blood agar, MacConkey, and Sabouraud Glucose agar medium (Oxoid, Hampshire, UK). All plates were incubated overnight at 37 °C. The Chocolate agar was maintained in the presence of CO_2_. After 24–48 h of incubation, each plate was examined and bacterial identification and antimicrobial susceptibility test were performed. The bacterial identification and antimicrobial susceptibility test were performed via technology Vitek 2 (bioMe’rieux, Marcy l’Etoile, France), following the manufacturer’s recommendations. The results of antimicrobial susceptibility were interpreted as “susceptible”, “resistant”, or “intermediate” according to EUCAST guidelines.

### 4.3. Ethical Consideration Statement

Ethical approval by the Human Research Ethics Committee was not requested. The present study used laboratory management data, collected from database. This is a retrospective study and not directly associated with patients.

### 4.4. Statistical Analysis

Demographic data of patients, including age, gender, isolated strain(s), and drug sensitivity results, were used for the analysis. The crude incidence and age- and sex-standardized incidence were calculated. The chi-framework test was used to compare the differences in the incidence of bacteria in hospitalized patients and the differences among antibiotic sensitivities over the range of years considered in the study. The *p*-value was calculated using the chi-squared test for a row-by-column contingency table with appropriate degrees of freedom. *p* < 0.05 was considered statistically significant. The IBM Statistical Package for Social Sciences Version 22.00 (SPSS Inc, Chicago, USA (http://www.spss.com)) was used for data analysis.

## 5. Conclusions

Hospital surveillance studies of blood infections allow for a deeper understanding of BSI-causing microorganisms and their pattern of antibiotic susceptibility to improve empirical antibiotic therapy [44]. It is essential to evaluate the etiological agents, the results of microbial culture, and antimicrobial susceptibility, in order to be able to follow the trend of resistance to the most frequently administered antibiotics [45,46]. Efficient control methods are needed to decrease resistance to antibiotic drugs and to ensure that patients receive effective treatment [47,48]. Therefore, programs should be implemented to improve the quality of empirical therapy in patients with suspected BSI and the optimization of definitive therapy, improving the antimicrobial management program in our university hospital [49].

## Figures and Tables

**Figure 1 antibiotics-09-00851-f001:**
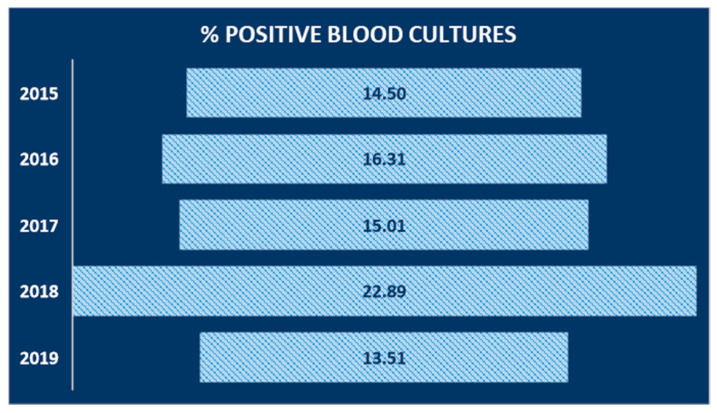
Incidence of bloodstream infection (BSI) cases by year of study, expressed as a percentage relative to the total number of positive cases out of the total number of cases present per year of study.

**Figure 2 antibiotics-09-00851-f002:**
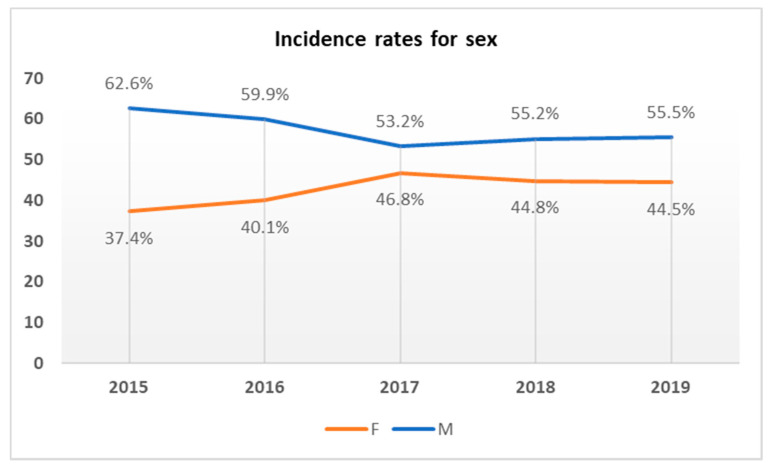
Incidence of BSI cases by study year associated with the sex of the patients involved.

**Figure 3 antibiotics-09-00851-f003:**
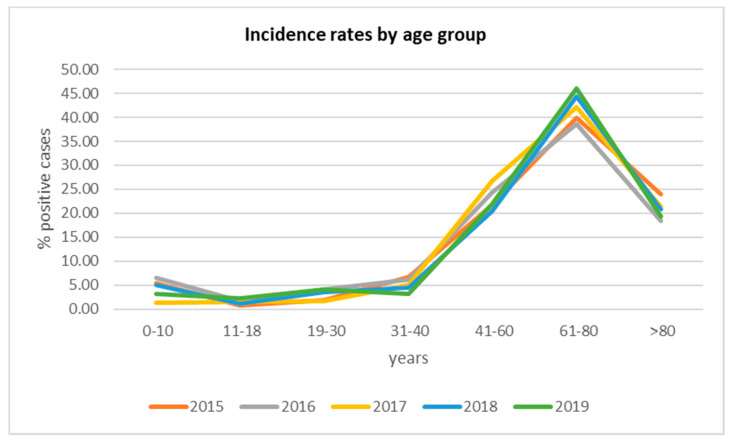
Distribution of positive cases by age group.

**Figure 4 antibiotics-09-00851-f004:**
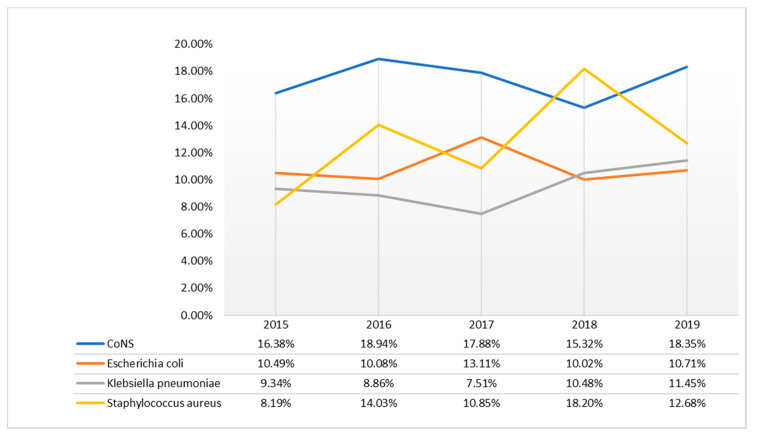
Trend in percentage of batteries caused by the most frequent isolates.

**Table 1 antibiotics-09-00851-t001:** Cases of bacteremia distributed by year.

Year	2015	2016	2017	2018	2019	Total
**Positive**	696	734	839	868	812	3949
**Negative**	4104	3766	4752	2924	5199	20,745
**Total**	4800	4500	5591	3792	6011	24,694

**Table 2 antibiotics-09-00851-t002:** Antimicrobial resistance profile of *Staphylococcus aureus* from patients with BSI to commonly used antibiotics.

*St. aureus*	2015		2016		2017		2018		2019	
Antibiotic	n. Assays	R %	n. Assays	R %	n. Assays	R %	n. Assays	R %	n. Assays	R %
**Fusidic acid**	60	0	103	1.9	91	5.5	158	4.4	103	4.9
**Clindamycin**	60	36.7	103	27.2	91	39.6	158	39.9	103	35.9
**Daptomycin**	60	0	103	1.0	91	5.5	158	5.1	103	3.9
**Erythromycin**	60	43.3	103	49.5	91	50.5	158	53.2	103	42.7
**Gentamicin**	60	13.3	103	13.6	91	9.9	158	10.8	103	7.8
**Levofloxacin**	57	33.3	79	35.4	77	53.2	116	41.4	90	27.8
**Linezolid**	60	0	103	0	91	4.4	158	4.4	103	2.9
**Oxacillin**	60	26.7	103	50.5	91	53.8	158	50.6	103	36.9
**Penicillin G**	60	86.7	102	85.3	91	84.6	158	86.7	103	68.9
**Rifampicin**	57	7.0	79	12.7	77	0	116	3.4	90	7.8
**Teicoplanin**	60	0	103	1.9	91	5.5	157	5.7	103	7.8
**Tetracycline**	60	9.6	103	9.8	91	8.8	158	3.2	103	9.8
**Tigecycline**	58	0	103	0	91	4.4	157	2.5	103	1.0
**Trimethoprim/** **Sulfam.**	60	1.7	103	2.0	91	5.5	157	1.9	103	4.9
**Vancomycin**	60	0	103	1.9	91	4.4	158	4.4	103	3.9

**Table 3 antibiotics-09-00851-t003:** Antimicrobial resistance profile of CoNS strains from patients with BSI to commonly used antibiotics.

CoNS	2015		2016		2017		2018		2019	
Antibiotic	n. Assays	R %	n. Assays	R %	n. Assays	R %	n. Assays	R %	n. Assays	R %
**Fusidic acid**	212	28.3	257	36.6	274	27.0	230	31.7	242	33.1
**Clindamycin**	216	62.5	257	57.2	272	55.1	232	54.3	243	51.4
**Daptomycin**	211	0.5	257	1.2	269	2.2	229	0.9	243	2.9
**Linezolid**	200	0	237	0	258	0.4	231	6.9	243	3.3
**Oxacillin**	206	81.1	233	77.3	249	72.7	229	78.6	243	77.8
**Rifampicin**	207	28.0	233	31.8	250	32.4	201	37.3	201	34.3
**Vancomycin**	216	0.5	257	0.4	273	1.1	232	0.4	242	3.7

**Table 4 antibiotics-09-00851-t004:** Antimicrobial resistance profile of *Klebsiella pneumoniae* from patients with BSI to commonly used antibiotics.

*Klebsiella pneumoniae*	2015		2016		2017		2018		2019	
Antibiotic	n. Assays	R %	n. Assays	R %	n. Assays	R %	n. Assays	R %	n. Assays	R %
**Amoxicillin/A. Clav.**	67	77.6	60	76.7	53	79.2	62	82.3	116	94.8
**Cefotaxime**	67	85.1	65	73.8	63	87.3	91	81.3	92	92.4
**Ceftazidime**	67	79.1	65	73.8	63	85.7	91	84.6	93	92.5
**Ciprofloxacin**	67	86.6	65	73.8	63	85.7	91	76.9	93	89.2
**Colistin**	66	21.2	63	27.0	55	16.4	75	12.3	88	26.1
**Ertapenem**	59	62.7	65	64.6	63	77.8	91	54.9	93	65.6
**Fosfomycin**	67	22.4	60	28.3	53	20.7	63	14.3	68	38.2
**Meropenem**	67	64.2	65	64.6	63	74.6	91	54.9	93	65.6
**Piperacillin/Tazobactam**	67	79.1	65	70.8	63	82.5	91	81.3	93	79.6
**Trimethoprim/Sulf.**	67	77.6	65	69.2	63	88.9	91	70.3	93	58.1

**Table 5 antibiotics-09-00851-t005:** Antimicrobial resistance profile of *Escherichia coli* from patients with BSI to commonly used antibiotics.

*Escherichia coli*	2015		2016		2017		2018		2019	
Antibiotic	n. Assays	R %	n. Assays	R %	n. Assays	R %	n. Assays	R %	n. Assays	R %
**Amoxicillin/A. Clav.**	70	50.0	66	56.1	103	39.8	84	51.2	110	68.2
**Cefotaxime**	73	53.5	74	54.1	110	52.7	87	57.5	83	54.2
**Ciprofloxacin**	73	61.6	74	68.9	110	66.3	87	67.8	87	71.3
**Ertapenem**	73	1.4	74	0	110	0	87	2.3	87	2.3
**Fosfomycin**	70	0	66	0	103	4.9	84	0	70	0
**Gentamicin**	73	34.2	74	40.5	110	27.3	87	27.6	87	31.0
**Imipenem**	73	0	74	0	110	0	74	0	23	0
**Meropenem**	73	0	74	0	110	0	87	2.3	87	2.2
**Piperacillin/tazobactam**	72	15.3	74	10.8	110	18.2	86	12.8	85	14.1
**Tigecycline**	72	0	73	0	109	0	85	0	81	1.2
**Trimethoprim/Sulf.**	73	41.1	74	43.2	110	45.5	87	52.9	87	55.2

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
