# Peer review of "Sepsis—A Retrospective Cohort Study of Bloodstream Infections"

_antibiotics, 2020, doi:10.3390/antibiotics9120851_

Round 1

Reviewer 1 Report

The authors have adequately addressed this reviewer's criticisms.

Author Response

Cover Letter in Response to Reviewer’s Comments

Name of journal: Antibiotics

Manuscript ID: antibiotics-989618

Title: Sepsis - a retrospective cohort study of bloodstream infections

Reviewer 1

The authors have adequately addressed this reviewer's criticisms.

Author Response to reviewer 1

  1. A. The authors have adequately addressed this reviewer's criticisms.
  2. R. Thank you for the revisions received and for the improvement made thanks to them.

Reviewer 2 Report

No additional comments.

The revised version may be accepted for publication in Antibiotics journal.

Author Response

Cover Letter in Response to Reviewer’s Comments

Name of journal: Antibiotics

Manuscript ID: antibiotics-989618

Title: Sepsis - a retrospective cohort study of bloodstream infections

Reviewer 2

No additional comments.

The revised version may be accepted for publication in Antibiotics journal.

Author Response to reviewer 2

  1. A. No additional comments. The revised version may be accepted for publication in Antibiotics journal.
  2. R. Thank you very much for the positive feedback received and the improvement made as a result of previous reviews.

Reviewer 3 Report

This manuscript is, " Sepsis - a retrospective cohort study of bloodstream infections " by Dr. Biagio Santella and his/her colleagues. This study is a cross-sectional, epidemiological report majorly investigating the distribution of the causative microorganisms and their susceptibilities. They found an increase in resistance to the latest generation of antibiotics over the years and thereby suggests an urgent need to improve the antimicrobial stewardship programs in the study hospital. However, numerous concerns were discovered as the following:

First, my leading concern was the novelty. Although the large patient population was collected in your work and this can offer sufficient data for the antimicrobial stewardship program, novel finding was not disclosed in your population.

Second, numerous definitions should be clarified. For an example, in your tittle, sepsis and bloodstream infection was different clinical events. What is the primary and secondary bacteremia? you should clarify these.

Third, to recognize the true bacteremia episode, patients with contaminate blood-culture sampling should be excluded. Therefore, in my opinion, many bacteremia episodes due to CoNS should be categorized as the contamination.

Fourth, to recognize useful information for the antimicrobial stewardship program, the overall patients should be categorized into community-onset and nosocomial-onset bacteremia.

Author Response

Dear Reviewer, Thank you very much for the comments and advice received regarding my article. Attached is the complete file with the answers.

Best regards,

Biagio Santella

Cover Letter in Response to Reviewer’s Comments

Name of journal: Antibiotics

Manuscript ID: antibiotics-989618

Title: Sepsis - a retrospective cohort study of bloodstream infections

Reviewer 3

This manuscript is, " Sepsis - a retrospective cohort study of bloodstream infections " by Dr. Biagio Santella and his/her colleagues. This study is a cross-sectional, epidemiological report majorly investigating the distribution of the causative microorganisms and their susceptibilities. They found an increase in resistance to the latest generation of antibiotics over the years and thereby suggests an urgent need to improve the antimicrobial stewardship programs in the study hospital. However, numerous concerns were discovered as the following:

First, my leading concern was the novelty. Although the large patient population was collected in your work and this can offer sufficient data for the antimicrobial stewardship program, novel finding was not disclosed in your population.

Second, numerous definitions should be clarified. For an example, in your tittle, sepsis and bloodstream infection was different clinical events. What is the primary and secondary bacteremia? you should clarify these.

Third, to recognize the true bacteremia episode, patients with contaminate blood-culture sampling should be excluded. Therefore, in my opinion, many bacteremia episodes due to CoNS should be categorized as the contamination.

Fourth, to recognize useful information for the antimicrobial stewardship program, the overall patients should be categorized into community-onset and nosocomial-onset bacteremia.

Author Response to reviewer 3

  1. A. First, my leading concern was the novelty. Although the large patient population was collected in your work and this can offer sufficient data for the antimicrobial stewardship program, novel finding was not disclosed in your population.

1.R. Thanks for the comment. The objective of this work, carried out for the first time at this hospital, has disclosed some data necessary to be able to establish a new antimicrobial management program. Although some results are not different from those present in other similar works, others show the presence of new strains resistant to some antibiotics recently introduced in the clinical setting, at this hospital; such as meropenem-resistant E. coli strains present in the last two years of study. These data should be followed and monitored to avoid the phenomenon of MDR, which is constantly growing and increasingly difficult to extinguish.

2.A. Second, numerous definitions should be clarified. For an example, in your tittle, sepsis and bloodstream infection was different clinical events. What is the primary and secondary bacteremia? you should clarify these.

2.R. Thank you for your suggestion. The definitions examined have been amply clarified and corrected in the various sections of the manuscript following your revision

3.A. Third, to recognize the true bacteremia episode, patients with contaminate blood-culture sampling should be excluded. Therefore, in my opinion, many bacteremia episodes due to CoNS should be categorized as the contamination.

3.R. Thanks for the comment. Regarding the CoNS strains as possible contaminants, it was disclosed in the manuscript, in the section of materials and methods, in the discussions and in the results, the procedure carried out upstream and the analysis carried out downstream to be able to exclude and differentiate the CoNS as true authors of the infection by those contaminants.

4.A. Fourth, to recognize useful information for the antimicrobial stewardship program, the overall patients should be categorized into community-onset and nosocomial-onset bacteremia.

4.R. Thanks for the comment. In order to distinguish the two types of bacteremia, I would need to know precisely the time of hospitalization of all patients in this study. Unfortunately, the analyzed databases do not report the exact time of hospitalization of patients in the various wards affiliated with the hospital. However, the data collected in this study are needed to initiate a new antimicrobial management program, pending further data entry to make a future work more complete and satisfying.

Reviewer 4 Report

Articles, like this, emphasizing the worsening situation of nosocomial diseases, including bloodstream infections and the consequent widespread of antimicrobial resistance, are extremely important to be.

The first sentence of the abstarct needs modification, as bloodstream infections are not the main, just among the top causes of morbidity and mortality in the world due to infectious diseases. One of the first sentences of introduction states that "BSIs affects approximately 30 million people, causing 6 million deaths each year in the world", howere the cited reference does not say anything about this.

The Materials and methods should be section 4, not section 2, according to "Instructions for authors".

"The mean of patients with BSIs was 16,4%." sentence appears 2 time in section 3.1.

The titles of figure 1 and 2 should be completed.

The abbreviation of CoNS should appear in the text, as well, not just in the abstarct.

"Out of 24.694 blood samples, BSIs were diagnosed to 3.949
209 patients." appears two times in the discussion section.

There is no need to mention in the discussion that antimicrobial resistance levels are shown in which tables (in the results section).

What can be the reason for the higher incidence rate of bloodstream infections in 2018, compared to other years?

In discussion, in lines 217-248, thea authors mention several  (new, not mentioned before) data, that are actually belong to results section. In the discussion it would be better just summarizing them and comparing to findings of other publications.

Is there any location-specific (University Hospital "San 89 Giovanni di Dio e Ruggi d'Aragona) reason that can explain the experienced  trends/incidence/resistance rates? Is there any plan for improving the findings at that hospital? What can be done theoretically, is there any good practice what other publications mention?

Author Response

Dear Reviewer, Thank you very much for the comments and advice received regarding my article. Attached is the complete file with the answers.

Best regards,

Biagio Santella

Cover Letter in Response to Reviewer’s Comments

Name of journal: Antibiotics

Manuscript ID: antibiotics-989618

Title: Sepsis - a retrospective cohort study of bloodstream infections

Reviewer 4

Articles, like this, emphasizing the worsening situation of nosocomial diseases, including bloodstream infections and the consequent widespread of antimicrobial resistance, are extremely important to be.

The first sentence of the abstarct needs modification, as bloodstream infections are not the main, just among the top causes of morbidity and mortality in the world due to infectious diseases. One of the first sentences of introduction states that "BSIs affects approximately 30 million people, causing 6 million deaths each year in the world", howere the cited reference does not say anything about this.

The Materials and methods should be section 4, not section 2, according to "Instructions for authors".

"The mean of patients with BSIs was 16,4%." sentence appears 2 time in section 3.1.

The titles of figure 1 and 2 should be completed.

The abbreviation of CoNS should appear in the text, as well, not just in the abstarct.

"Out of 24.694 blood samples, BSIs were diagnosed to 3.949

209 patients." appears two times in the discussion section.

There is no need to mention in the discussion that antimicrobial resistance levels are shown in which tables (in the results section).

What can be the reason for the higher incidence rate of bloodstream infections in 2018, compared to other years?

In discussion, in lines 217-248, thea authors mention several  (new, not mentioned before) data, that are actually belong to results section. In the discussion it would be better just summarizing them and comparing to findings of other publications.

Is there any location-specific (University Hospital "San 89 Giovanni di Dio e Ruggi d'Aragona) reason that can explain the experienced  trends/incidence/resistance rates? Is there any plan for improving the findings at that hospital? What can be done theoretically, is there any good practice what other publications mention?

Author Response to reviewer 4

  1. A. The first sentence of the abstarct needs modification, as bloodstream infections are not the main, just among the top causes of morbidity and mortality in the world due to infectious diseases. One of the first sentences of introduction states that "BSIs affects approximately 30 million people, causing 6 million deaths each year in the world", howere the cited reference does not say anything about this.

1.R. The first sentence of the abstract has been changed as suggested: " Bloodstream infections (BSI) are among the leading causes of morbidity and mortality worldwide, among infectious diseases". In addition, an appropriate reference (Fleischmann C, Scherag A, Adhikari NKJ, Hartog CS, Tsaganos T, Schlattmann P, et al. Assessment of Global Incidence and Mortality of Hospital-treated Sepsis. Current Estimates and Limitations. American Journal of Respiratory and Critical Care Medicine. 2016;193(3):259-72.) has been added to the introductory sentence: "BSI affects approximately 30 million people, causing 6 million deaths worldwide each year” [2].

2.A. The Materials and methods should be section 4, not section 2, according to "Instructions for authors".

2.R. Thanks for your attention, we have made the relevant correction. We also followed the journal's "author recommendations" more carefully.

3.A. "The mean of patients with BSIs was 16,4%." sentence appears 2 time in section 3.1.

3.R. The copy of this sentence has been deleted.

4.A. The titles of figure 1 and 2 should be completed.

4.R. The title of figure 1 has been completed as follows: Incidence of BSI cases by year of study, expressed as a percentage relative to the total number of positive cases out of the total number of cases present per year of study. The title of figure 2 has been completed as follows: Incidence of BSI cases by study year associated with the sex of the patients involved.

5.A. The abbreviation of CoNS should appear in the text, as well, not just in the abstarct.

5.R. The CoNS abbreviation has been inserted in the text in line 61.

6.A. "Out of 24.694 blood samples, BSIs were diagnosed to 3.949 patients." appears two times in the discussion section.

6.R. This error has been corrected.

7.A. There is no need to mention in the discussion that antimicrobial resistance levels are shown in which tables (in the results section).

7.R. References to tables in the discussion section have been removed

8.A. What can be the reason for the higher incidence rate of bloodstream infections in 2018, compared to other years?

8.R. The answer to this question was included in the discussion, and it is the following: "A fairly linear incidence trend (15-16%) was calculated for the years under review, with the exception of 2018, where an incidence exceeds 22%. This higher value can be justified by a lower total number of blood cultures received and by a high positivity, probably caused by the S. aureus strains, more frequent in the year under review ".

9.A. In discussion, in lines 217-248, thea authors mention several  (new, not mentioned before) data, that are actually belong to results section. In the discussion it would be better just summarizing them and comparing to findings of other publications.

9.R. The discussion section has been completely revised on the above comment. Some data from the discussions have been moved to the results. The data has been summarized and better discussed in the appropriate section as suggested. They were compared with other studies and other citations added on the matter.

10.A. Is there any location-specific (University Hospital "San 89 Giovanni di Dio e Ruggi d'Aragona) reason that can explain the experienced  trends/incidence/resistance rates? Is there any plan for improving the findings at that hospital? What can be done theoretically, is there any good practice what other publications mention?

10.R. I am sorry to answer you, that despite having searched the literature, I have not found any location specific reason that would justify the data shown in this study. One plan to improve these results could be to follow the epidemiological studies carried out at the facility and use them as new guidelines for the treatment of BSI, paying close attention to which antibiotic to use in the treatment of these infections. Comment present in the conclusions section.

Round 2

Reviewer 3 Report

My concerns, in terms of the novelty, definitions, and contaminant issues, were not substantially improved in the revised manuscript.